# Inter-Satellite Cooperative Offloading Decision and Resource Allocation in Mobile Edge Computing-Enabled Satellite–Terrestrial Networks

**DOI:** 10.3390/s23020668

**Published:** 2023-01-06

**Authors:** Minglei Tong, Song Li, Xiaoxiang Wang, Peng Wei

**Affiliations:** 1School of Information and Communication Engineering, Beijing University of Posts and Telecommunications, Beijing 100876, China; 2Key Laboratory of Universal Wireless Communications, Ministry of Education, Beijing University of Posts and Telecommunications, Beijing 100876, China; 3School of Information and Control Engineering, China University of Mining and Technology, Xuzhou 221116, China

**Keywords:** mobile edge computing, satellite–terrestrial networks, inter-satellite cooperation, offloading decision, resource allocation

## Abstract

Mobile edge computing (MEC)-enabled satellite–terrestrial networks (STNs) can provide task computing services for Internet of Things (IoT) devices. However, since some applications’ tasks require huge amounts of computing resources, sometimes the computing resources of a local satellite’s MEC server are insufficient, but the computing resources of neighboring satellites’ MEC servers are redundant. Therefore, we investigated inter-satellite cooperation in MEC-enabled STNs. First, we designed a system model of the MEC-enabled STN architecture, where the local satellite and the neighboring satellites assist IoT devices in computing tasks through inter-satellite cooperation. The local satellite migrates some tasks to the neighboring satellites to utilize their idle resources. Next, the task completion delay minimization problem for all IoT devices is formulated and decomposed. Then, we propose an inter-satellite cooperative joint offloading decision and resource allocation optimization scheme, which consists of a task offloading decision algorithm based on the Grey Wolf Optimizer (GWO) algorithm and a computing resource allocation algorithm based on the Lagrange multiplier method. The optimal solution is obtained by continuous iterations. Finally, simulation results demonstrate that the proposed scheme achieves relatively better performance than other baseline schemes.

## 1. Introduction

With the development of space technology and communication technology, the number of satellites in space has increased dramatically. In the future 6G network, satellite networks can be used as supplements to terrestrial cellular networks to achieve seamless full-domain coverage [1]. In addition, the satellite mobile system also has high viability when disasters happen [2]. When terrestrial network entities are destroyed by natural disasters, such as earthquakes and typhoons, people can rely on satellites to access services. Therefore, for remote areas that are not covered by terrestrial networks or disaster areas where terrestrial networks are destroyed, satellite networks can replace or assist terrestrial networks to provide services to users. The integration of satellite networks and terrestrial networks, i.e., satellite–terrestrial networks (STNs), has received extensive attention from the academic world and the industrial world [3]. STNs have advantages such as wide coverage, strong robustness, and strong damage resistance.

Since the rise in popularity of smartphones and Internet of Things (IoT) devices, various applications have been emerging endlessly in people’s daily life. Some of them, such as interactive games, face recognition, and augmented reality, require large amounts of computing resources [4]. Due to the physical size limitation and energy supply constraints, the computing capacity of devices is limited [5]. IoT devices compute tasks of these applications locally, which significantly imposes delay and reduces the quality of the experience.

Mobile edge computing (MEC) provides information technology and cloud computing capabilities within the radio access networks, which are extremely close to users [6]. Edge servers are modular and miniaturized because they host cloud services that are scaled to the edge [7]. MEC servers can be deployed at multiple locations, such as base stations (BSs), access points, and satellites. Computation offloading technology makes it possible for users to utilize the computing capabilities at the edge of the network [8]. Due to MEC, the delay can be reduced and the quality of service can also be improved [9].

In this paper, we consider applying MEC technology to STNs, i.e., MEC-enabled STNs. Low earth orbit (LEO) satellite constellations can achieve seamless global coverage [10]. Moreover, LEO satellite networks have advantages of short satellite–terrestrial transmission distance and low construction cost. In addition, LEO satellite IoT can complement and extend terrestrial IoT to better achieve IoT applications, such as environmental monitoring, marine monitoring, and border control [11]. MEC servers can be deployed at LEO satellites to handle computing tasks from devices [12]. For remote areas with sparse users and insufficient terrestrial network coverage, such as mountainous areas and islands, offloading tasks to satellites is the only option. Compared to the computing resources of MEC servers deployed at BSs, the computing resources of MEC servers deployed at satellites are less due to the satellite payload constraints. Therefore, a single LEO satellite’s MEC server may sometimes be insufficient to assist all IoT devices in computing tasks.

For sparsely populated areas where IoT devices are widely spaced, such as deserts and oceans, inter-satellite cooperation is an effective coping approach. Sometimes, the computing workload is distributed unevenly in different space, which leads to the fact that some MEC servers may need to handle tasks more than their computing capabilities, whereas some other MEC servers still have surplus computing resources [13]. The LEO satellites launched recently are equipped with laser terminals, which can establish inter-satellite links (ISLs) with neighboring satellites [14]. For example, in the BeiDou system, ISLs can be established between two satellites if they are visible [15]. Similar to BS cooperation in terrestrial networks, multiple LEO satellites can also achieve cooperation, which compensates for the insufficient computing resources of a single LEO satellite’s MEC server.

The local satellite migrates some tasks to neighboring satellites, which can utilize the computing resources of neighboring satellites’ MEC servers to compute tasks, thus reducing the task completion delay. Compared with local computing, task offloading introduces additional overhead, such as transmission delay. Moreover, the computing resources of MEC servers are relatively limited compared with the central cloud. Therefore, it is very important to make task offloading decisions and allocate computing resources properly. According to whether tasks are divisible, computation offloading can be divided into two categories, namely binary offloading and partial offloading [16]. Binary offloading involves integer programming, so the corresponding task offloading problem is generally NP-hard. Furthermore, assigning computing resources to tasks of different devices complicates the problem and increases the difficulty of algorithm design [17].

In this paper, we consider an STN consisting of IoT devices, a local satellite, and neighboring satellites. The tasks of the IoT devices within the coverage area of the local satellite are considered, and the neighboring satellites are regarded as computing nodes without considering the tasks of the IoT devices within their coverage areas. Neighboring satellites assist the local satellite in computing IoT devices’ tasks by inter-satellite cooperation. We propose an inter-satellite cooperative task offloading and resource allocation scheme in an MEC-enabled STN, which minimizes the task completion delay for all IoT devices under the condition that the tasks are indivisible.

The main contributions of this paper are as follows:The system model of the MEC-enabled STN is established to provide MEC services in remote or disaster areas by utilizing inter-satellite cooperation. The joint task offloading and computing resource allocation problem is formulated to minimize the task completion delay and decomposed into a task offloading decision problem and a computing resource allocation problem.An inter-satellite cooperative joint offloading decision and resource allocation optimization scheme, which consists of a task offloading decision algorithm based on the Grey Wolf Optimizer (GWO) algorithm and a computing resource allocation algorithm based on the Lagrange multiplier method is proposed. The optimal task offloading decision and computing resource allocation are obtained by the proposed scheme.The performance of the proposed scheme is evaluated by comparing it with other baseline schemes with respect to the variation of some parameters. Simulation results demonstrate the performance gain of the proposed scheme.

The rest of this paper is organized as follows. Section 2 introduces related works. Section 3 describes the system model. The task completion delay minimization problem for all IoT devices is formulated in Section 4. In order to solve this problem, the corresponding algorithm is designed in Section 5. In Section 6, simulations and an analysis are presented. Section 7 concludes this paper.

## 2. Related Work

Non-terrestrial networks, particularly satellite networks, were proposed to be integrated with terrestrial networks in the early 2030s. Moreover, the most anticipated function of the STN was to provide coverage to rural and remote areas [18]. The authors in [19] proposed satellite MEC. MEC servers deployed at LEO satellites enabled devices without a nearby MEC server to enjoy MEC services through satellite links. In addition, a cooperative computation offloading model was designed to realize parallel computation in STNs, which minimized user-perceived delay and system energy consumption by optimizing task scheduling. Through radio frequency (RF) communication or optical communication, ISLs can realize data communication, interconnection, and information relay. Moreover, compared with RF ISLs, optical ISLs have the advantages of broader bandwidth, lower energy consumption, and lower interference [20]. According to [21], inter-satellite laser technologies can establish ISLs and achieve transmission between satellites at a high speed. With the aid of optical or visible light communication systems, LEO satellites can achieve connection through intra-plane ISLs and inter-plane ISLs [22]. In [23], some satellites temporarily formed a task group, where the access satellite was responsible for task and resource scheduling, while others were responsible for computing tasks. The authors in [24] proposed a double-edge computing STN architecture, where an access satellite can send tasks to other satellites for computing. They also proposed a double-edge computation offloading algorithm to minimize system energy consumption and reduce task offloading delay. In [25], a double-edge intelligent integrated STN was designed. Satellite MEC servers were grouped by ISLs, and they can assist terrestrial MEC servers in computing tasks. A task migration strategy based on a greedy algorithm was proposed to achieve load balancing and reduce system delay. The authors in [26] proposed a hybrid cloud and edge computing LEO satellite network architecture, where the tasks of ground users can be completed in one of three places, i.e., the ground user themselves, the LEO satellite, or the terrestrial cloud server. They also came up with a distributed algorithm to minimize the sum energy consumption of the ground users. The authors in [27] proposed an MEC framework for the terrestrial–satellite IoT. With the assistance of terrestrial–satellite terminals, IoT mobile devices can offload tasks to LEO satellites for computing. They also proposed an energy-efficient algorithm for computation offloading and resource allocation, thereby minimizing the weighted sum of energy consumption of IoT mobile devices.

Most of the above works investigate the satellite–terrestrial cooperation or the inter-satellite cooperation in the presence of terrestrial networks, where satellites usually assist terrestrial entities. However, there are few works that investigate inter-satellite cooperation to assist users in computing tasks in areas not covered by terrestrial networks.

Algorithms for computation offloading include heuristic algorithms, meta-heuristic algorithms, game theory-based algorithms, etc. [28]. Algorithms for resource allocation include convex optimization algorithms, heuristic algorithms, game theory-based algorithms, etc. Convex optimization algorithms are widely investigated and can easily obtain suboptimal solutions. However, they are complex and not easy to implement in actual systems. Heuristic algorithms are efficient and can quickly obtain solutions, but they are prone to fall into the local optimum. Meta-heuristic algorithms have too many parameters, which are not easy to adjust. Game theory-based algorithms are simple, flexible, and easy to implement. However, the obtained solution may not be globally optimal and the Nash equilibrium needs to be reached through continuous iterations [29]. The GWO algorithm is a meta-heuristic algorithm with few parameters. Therefore, we propose a task offloading decision algorithm based on the GWO algorithm.

## 3. System Model

In this section, the network model is proposed first. Then, the channel model is presented. Lastly, the computation model is introduced.

### 3.1. Network Model

As shown in Figure 1, a remote area beyond the reach of terrestrial BSs is considered in this paper. There is an LEO satellite 1 in space, called the local satellite, around which there are M−1 neighboring satellites, i.e., LEO satellite *j*, j∈M, where $M=\left\{2,3,\ldots ,M\right\}$. All satellites in the same orbit move in the same circular direction. We define a satellite within the communication range of the local satellite at the moment as a neighboring satellite. These LEO satellites are all deployed with MEC servers. *N* IoT devices are randomly distributed in the ground area covered by the local satellite, and the set composed of them is denoted by N. These IoT devices are equipped with satellite antennas and can communicate with the local satellite directly. Each IoT device requests an indivisible task; for instance, IoT device *i* requests task *i*, denoted by $\left(c_{i},a_{i}\right)$, where $c_{i}$ is the computation intensity, i.e., computing resources required for computing one bit of task *i* (cycles per bit), and $a_{i}$ is the data size of task *i* (bits). Different IoT devices have different computing capabilities. MEC servers have more computing resources than IoT devices, but the computing resources of MEC servers are still limited. Task generation mechanisms may be periodic generation, random generation, and event-triggered generation. The problem we study is to build the corresponding task model for the set of tasks generated by the IoT devices in the network at a certain moment. We assume that each IoT device has only one task at the moment and perform computation offloading according to the task model. Based on the offloading decision, the local computing portion of the task of each IoT device is computed by the IoT device itself. The satellite allocates its computing resources to the offloading portion of the tasks of the IoT devices and concurrently computes the tasks offloaded by the IoT devices. There are three strategies to complete task *i*, i∈N. IoT device *i* computes task *i* locally, i.e., local computing. IoT device *i* offloads task *i* to the local satellite, and task *i* is computed by the local satellite’s MEC server, i.e., local satellite edge computing. IoT device *i* offloads task *i* to the local satellite, then task *i* is migrated to neighboring satellite *j*, j∈M and computed by the neighboring satellite’s MEC server, i.e., neighboring satellite edge computing.

### 3.2. Channel Model

In order to avoid the interference among IoT devices, orthogonal frequency division multiple access (OFDMA) technology is used. Subcarriers with equal bandwidth are allocated to IoT devices. The satellite–terrestrial transmission links between IoT devices and the local satellite use the Ka band [27] and have high communication rates. Since the line-of-sight (LOS) transmission is mainly considered in the link, the Rice channel is used. Free-space path loss (FSPL) and additive white Gaussian noise (AWGN) are taken into consideration. The achievable data rate of the link between IoT device *i* and the local satellite is: (1)$$r_{i,1}=B_{i,1}\log_{2}\left(1+\frac{p_{i}g_{i}H_{i,1}}{N_{0_{i,1}}}\right),$$
where $B_{i,1}$ is the subchannel bandwidth, $p_{i}$ is the transmission power of IoT device *i*, $g_{i}$ is the antenna gain of IoT device *i*, $H_{i,1}$ is the channel gain of the link, and $N_{0_{i,1}}$ is the noise power of the link. $H_{i,1}={\left(h_{i,1}\right)}^{2}$, where the channel coefficient $h_{i,1}=\rho_{i,1}/{\left(L_{i,1}^{f}\right)}^{1/2}$, $\rho_{i,1}$ is the independent identically distributed Rice stochastic variable with a unit variance, $L_{i,1}^{f}$ is FSPL, $L_{i,1}^{f}={\left(4\pi d_{i,1}f/c\right)}^{2}$, *c* is the light velocity (meter per second), $d_{i,1}$ is the distance between IoT device *i* and the local satellite (meter), and *f* is the carrier frequency (hertz). $d_{i,1}^{\min}=h_{s}$, as Figure 2 depicted, $d_{i,1}^{\max}={\left[{\left(R_{e}\right)}^{2}+{\left(R_{e}+h_{s}\right)}^{2}-2R_{e}\left(R_{e}+h_{s}\right)\cos\theta_{\mathrm{ec}}\right]}^{1/2}$, where $R_{e}$ is the earth radius, $h_{s}$ is the LEO satellite orbit altitude, and $\theta_{\mathrm{ec}}$ is the covering geocentric angle. $\theta_{\mathrm{ec}}=\arccos\left[R_{e}\cos\theta_{u}^{\min}/\left(R_{e}+h_{s}\right)\right]-\theta_{u}^{\min}$, $\theta_{u}^{\min}$ is the minimum elevation angle of the IoT devices on the ground to the satellite [30].

The ISLs between the local satellite and neighboring satellites use the Ka band [31]. FSPL and AWGN are taken into account. The achievable data rate of the link between the local satellite and neighboring satellite *j*, j∈M is: (2)$$r_{1,j}=B_{1,j}\log_{2}\left(1+\frac{PGH_{1,j}}{N_{0_{1,j}}}\right),$$
where $B_{1,j}$ is the subchannel bandwidth, *P* is the transmission power of the local satellite, *G* is the antenna gain of the local satellite, $H_{1,j}$ is the channel gain of the link, and $N_{0_{1,j}}$ is the noise power of the link. $G\approx {\left(4D_{\mathrm{TA}}f/c\right)}^{2}$, and $D_{\mathrm{TA}}$ is the diameter of the local satellite’s transmitting antenna [31]. $H_{1,j}={\left(h_{1,j}\right)}^{2}$, where the channel coefficient $h_{1,j}=1/{\left(L_{1,j}^{f}\right)}^{1/2}$, $L_{1,j}^{f}$ is FSPL, $L_{1,j}^{f}={\left(4\pi l_{1,j}f/c\right)}^{2}$, and $l_{1,j}$ is the distance between the local satellite and neighboring satellite *j* (meter).

Normally, each satellite has four neighboring satellites, two in the same orbit and the other two in the left and right orbits. The distance between the local satellite and the neighboring satellite in the same orbit is $l_{1,j}=\sqrt{2}R\sqrt{1-\cos\left(360^{{}^{\circ }}/J\right)}$, where *R* is the orbital radius, $R=h_{s}+R_{e}$, and *J* is the number of satellites in each orbit. The distance between the local satellite and the neighboring satellite in different orbits is $l_{1,j}=\omega \cos\left(lat\right)$, $\omega =\sqrt{2}R\sqrt{1-\cos\left(360^{{}^{\circ }}/2I\right)}$, where lat is the latitude where the ISL of the current satellite locate, and *I* is the number of orbits of satellite constellation [32].

### 3.3. Computation Model

To represent the task offloading strategies, we set indicator variables, i.e., $y_{i,0}$, $y_{i,1}$, $y_{i,j}\in \left\{0,1\right\}$, ∀i∈N, and ∀j∈M. When task *i* is completed by local computing, $y_{i,0}=1$, otherwise, $y_{i,0}=0$. When task *i* is completed by local satellite edge computing, $y_{i,1}=1$, otherwise, $y_{i,1}=0$. When task *i* is completed by neighboring satellite edge computing, $y_{i,j}=1$, otherwise, $y_{i,j}=0$.

When $y_{i,0}=1$, the task computing delay of IoT device *i* is: (3)$$D_{i,0}=\frac{c_{i}a_{i}}{f_{i,0}},$$
where $f_{i,0}$ is the computing capacity of IoT device *i* (cycles per second).

When $y_{i,1}=1$, the task completion delay of IoT device *i* includes the delay with which IoT device *i* offloads task *i* to the local satellite, the propagation delay between IoT device *i* and the local satellite, and the delay that the local satellite’s MEC server computes task *i*.
(4)$$D_{i,1}=\frac{a_{i}}{r_{i,1}}+\frac{2d_{i,1}}{c}+\frac{c_{i}a_{i}}{f_{i,1}},$$
where $f_{i,1}$ is the computing resource allocated to task *i* by MEC server of the local satellite (cycles per second).

When $y_{i,j}=1$, the task completion delay of IoT device *i* includes the delay that IoT device *i* offloads task *i* to the local satellite, the propagation delay between IoT device *i* and the local satellite, the delay with which the local satellite migrates task *i* to neighboring satellite *j*, the propagation delay between the local satellite and neighboring satellite *j*, and the delay with which neighboring satellite *j*’s MEC server computes task *i*.
(5)$$D_{i,j}=\frac{a_{i}}{r_{i,1}}+\frac{2d_{i,1}}{c}+\frac{a_{i}}{r_{1,j}}+\frac{2l_{1,j}}{c}+\frac{c_{i}a_{i}}{f_{i,j}},$$
where $f_{i,j}$ is the computing resource allocated to task *i* by the MEC server of neighboring satellite *j* (cycles per second).

In all of the cases above, the delay of IoT devices downloading task computing results from the local satellite or neighboring satellites is neglected, because the size of output resulting data obtained after task computation is usually much smaller than the size of input data [33].

## 4. Problem Formulation

The task completion delay of all IoT devices can be expressed as:(6)$$\begin{array}{cc} U & =\sum_{i\in N}\left(y_{i,0}D_{i,0}+y_{i,1}D_{i,1}+\sum_{j\in M}y_{i,j}D_{i,j}\right)\\ \; & =\sum_{i\in N}[y_{i,0}\frac{c_{i}a_{i}}{f_{i,0}}+y_{i,1}\left(\frac{a_{i}}{r_{i,1}}+\frac{2d_{i,1}}{c}+\frac{c_{i}a_{i}}{f_{i,1}}\right)+\sum_{j\in M}y_{i,j}\left(\frac{a_{i}}{r_{i,1}}+\frac{2d_{i,1}}{c}+\frac{a_{i}}{r_{1,j}}+\frac{2l_{1,j}}{c}+\frac{c_{i}a_{i}}{f_{i,j}}\right)]. \end{array}$$

The optimization objective of this paper is minimizing the task completion delay of all IoT devices. The optimization variables are $y_{i,0}$, $y_{i,1}$, $y_{i,j}$, $f_{i,1}$, and $f_{i,j}$. The problem P is as follows.
(7a)$$\begin{array}{cc} P: & \min_{\left\{\begin{array}{c} y_{i,0},y_{i,1},y_{i,j},\\ f_{i,1},f_{i,j} \end{array}\right\}}\;U \end{array}$$
(7b)$$\begin{array}{cc} \; & s.t.\;\sum_{i\in N}y_{i,1}f_{i,1}\leq F_{1}, \end{array}$$
(7c)$$\begin{array}{cc} \; & \sum_{i\in N}y_{i,j}f_{i,j}\leq F_{j},\forall j\in M, \end{array}$$
(7d)$$\begin{array}{cc} \; & 0\leq f_{i,1}\leq F_{1},\forall i\in N, \end{array}$$
(7e)$$\begin{array}{cc} \; & 0\leq f_{i,j}\leq F_{j},\forall i\in N,\forall j\in M, \end{array}$$
(7f)$$\begin{array}{cc} \; & y_{i,0}+y_{i,1}+\sum_{j\in M}y_{i,j}=1,\forall i\in N, \end{array}$$
(7g)$$\begin{array}{cc} \; & y_{i,0},y_{i,1},y_{i,j}\in \left\{0,1\right\},\forall i\in N,\forall j\in M. \end{array}$$

Equation ([Disp-formula FD7a-sensors-23-00668]) is the objective function. Equation ([Disp-formula FD7b-sensors-23-00668]) represents that computing resources allocated to task *i* cannot surpass the resources owned by the local satellite’s MEC server, i.e., $F_{1}$. Equation ([Disp-formula FD7c-sensors-23-00668]) shows that computing resources allocated to task *i* cannot exceed the resources of the MEC server of neighboring satellite *j*, i.e., $F_{j}$. Equations ([Disp-formula FD7d-sensors-23-00668]) and ([Disp-formula FD7e-sensors-23-00668]) mean that the MEC server’s computing resource allocation for task *i* is non-negative. Equation ([Disp-formula FD7f-sensors-23-00668]) denotes that only one strategy is selected to complete task *i*. Equation ([Disp-formula FD7g-sensors-23-00668]) means that $y_{i,0}$, $y_{i,1}$, and $y_{i,j}$ are 0 or 1.

P consists of the IoT devices’ task offloading decision and the computing resource allocation of the local satellite’s MEC server and neighboring satellites’ MEC servers. P includes discrete variables and continuous variables, so it is a mixed-integer nonlinear programming problem (MINLP), which is NP-hard.

## 5. Algorithm Design

From P, it can be seen that $y_{i,1}$ and $f_{i,1}$, $y_{i,j}$ and $f_{i,j}$ interact with each other, respectively. Since the variables are coupled, it is necessary to decouple them. Assuming $y_{i,0}^{*}$, $y_{i,1}^{*}$, and $y_{i,j}^{*}$, ∀i∈N, ∀j∈M are given, the sets $N^{D}=\left\{i|y_{i,0}^{*}=1\right\}$, $N^{L}=\left\{i|y_{i,1}^{*}=1\right\}$, and $N^{N}=\left\{i|y_{i,j}^{*}=1,\forall j\in M\right\}$ can be obtained. P can be transformed into P1.
(8a)$$\begin{array}{cc} P1: & \min_{\left\{f_{i,1},f_{i,j}\right\}}\;\sum_{i\in N^{D}}\frac{c_{i}a_{i}}{f_{i,0}}+\sum_{i\in N^{L}}\left(\frac{a_{i}}{r_{i,1}}+\frac{2d_{i,1}}{c}+\frac{c_{i}a_{i}}{f_{i,1}}\right)+\sum_{i\in N^{N}}\sum_{j\in M}\left(\frac{a_{i}}{r_{i,1}}+\frac{2d_{i,1}}{c}+\frac{a_{i}}{r_{1,j}}+\frac{2l_{1,j}}{c}+\frac{c_{i}a_{i}}{f_{i,j}}\right), \end{array}$$
(8b)$$\begin{array}{cc} \; & s.t.\;\sum_{i\in N^{L}}f_{i,1}\leq F_{1}, \end{array}$$
(8c)$$\begin{array}{cc} \; & \sum_{i\in N^{N}}f_{i,j}\leq F_{j},\forall j\in M, \end{array}$$
(8d)$$\begin{array}{cc} \; & 0<f_{i,1}\leq F_{1},\forall i\in N^{L}, \end{array}$$
(8e)$$\begin{array}{cc} \; & 0<f_{i,j}\leq F_{j},\forall i\in N^{N},\forall j\in M. \end{array}$$

Equation ([Disp-formula FD8a-sensors-23-00668]) is the objective function. The meanings of ([Disp-formula FD8b-sensors-23-00668])–([Disp-formula FD8e-sensors-23-00668]) are similar to the meanings of ([Disp-formula FD7b-sensors-23-00668])–([Disp-formula FD7e-sensors-23-00668]). From P1, it can be seen that the delay of local computing is a constant. Moreover, the computing resource allocation problem of the local satellite’s MEC server and the computing resource allocation problem of the neighboring satellites’ MEC servers are independent of each other. Therefore, P1 can be decomposed into two sub-problems, i.e., P1.1 and P1.2, and then $f_{i,1}^{*}$ and $f_{i,j}^{*}$ can be obtained by solving P1.1 and P1.2, respectively.

P1.1 is the computing resource allocation problem for the local satellite’s MEC server, which is represented as: (9)$$\begin{array}{cc} P1.1: & \min_{\left\{f_{i,1}\right\}}\sum_{i\in N^{L}}\left(\frac{a_{i}}{r_{i,1}}+\frac{2d_{i,1}}{c}+\frac{c_{i}a_{i}}{f_{i,1}}\right),\\ \; & s.t.\;(8b)\;\mathrm{and}\;(8d). \end{array}$$

The objective function in (Equation 9) is set as $U^{L}$. Then we can have $\partial U^{L}/\partial f_{i,1}={-c}_{i}a_{i}/{\left(f_{i,1}\right)}^{2}$, which is less than 0. $\partial^{2}U^{L}/\partial {\left(f_{i,1}\right)}^{2}=2c_{i}a_{i}/{\left(f_{i,1}\right)}^{3}$, which is greater than 0. $\partial^{2}U^{L}/\left(\partial f_{i,1}\partial f_{j,1}\right)=0$, $i,j\in N^{L}$, i≠j. The Hessian matrix of $U^{L}$ is positive definite, so $U^{L}$ is convex. P1.1 can be solved by using the Lagrange multiplier method. The Lagrangian function is constructed as: (10)$$L\left(f_{i,1},\theta \right)=\sum_{i\in N^{L}}\left(\frac{a_{i}}{r_{i,1}}+\frac{2d_{i,1}}{c}+\frac{c_{i}a_{i}}{f_{i,1}}\right)+\theta \left(\sum_{i\in N^{L}}f_{i,1}-F_{1}\right),$$
where θ is the Lagrange multiplier and θ≥0. The KKT conditions of P1.1 are listed as: (11)$$\begin{array}{c} \frac{\partial L\left(f_{i,1},\theta \right)}{\partial f_{i,1}}=-\frac{c_{i}a_{i}}{{\left(f_{i,1}\right)}^{2}}+\theta =0, \end{array}$$(12)$$\begin{array}{c} \frac{\partial L\left(f_{i,1},\theta \right)}{\partial \theta }=\sum_{i\in N^{L}}f_{i,1}-F_{1}=0, \end{array}$$(13)$$\begin{array}{c} \theta \left(\sum_{i\in N^{L}}f_{i,1}-F_{1}\right)=0,\\ (8b)\;\mathrm{and}\;(8d), \end{array}$$(14)$$\begin{array}{c} \theta \geq 0. \end{array}$$

$f_{i,1}=\sqrt{c_{i}a_{i}/\theta }$ is obtained from (Equation 11). Substituting it into (Equation 12), we can obtain $\sum_{i\in N^{L}}\sqrt{c_{i}a_{i}/\theta }-F_{1}=0$, i.e., $\sqrt{\theta }=\left(\sum_{i\in N^{L}}\sqrt{c_{i}a_{i}}\right)/F_{1}$. Thus, $f_{i,1}=F_{1}\sqrt{c_{i}a_{i}}/\left(\sum_{i\in N^{L}}\sqrt{c_{i}a_{i}}\right)$. Considering (8d), we can deduce:(15)$$f_{i,1}^{*}=\min\left[\max\left(\frac{F_{1}\sqrt{c_{i}a_{i}}}{\sum_{i\in N^{L}}\sqrt{c_{i}a_{i}}},0\right),F_{1}\right].$$

P1.2 is the computing resource allocation problem for neighboring satellites’ MEC servers, which is expressed as: (16)$$\begin{array}{cc} P1.2: & \min_{\left\{f_{i,j}\right\}}\sum_{i\in N^{N}}\sum_{j\in M}\left(\frac{a_{i}}{r_{i,1}}+\frac{2d_{i,1}}{c}+\frac{a_{i}}{r_{1,j}}+\frac{2l_{1,j}}{c}+\frac{c_{i}a_{i}}{f_{i,j}}\right),\\ \; & s.t.\;(8c)\;\mathrm{and}\;(8e). \end{array}$$

Since the computing resource allocation of each neighboring satellite is independent of each other, P1.2 can be decomposed into M−1 sub-problems, one of which is P1.3.
(17)$$\begin{array}{cc} P1.3: & \min_{\left\{f_{i,j}\right\}}\sum_{i\in N^{N}}\left(\frac{a_{i}}{r_{i,1}}+\frac{2d_{i,1}}{c}+\frac{a_{i}}{r_{1,j}}+\frac{2l_{1,j}}{c}+\frac{c_{i}a_{i}}{f_{i,j}}\right),\\ \; & s.t.\;(8c)\;\mathrm{and}\;(8e). \end{array}$$

The objective function in (Equation 17) is set as $U^{N}$. Then, we can have $\partial U^{N}/\partial f_{i,j}={-c}_{i}a_{i}/{\left(f_{i,j}\right)}^{2}$, which is less than 0. $\partial^{2}U^{N}/\partial {\left(f_{i,j}\right)}^{2}=2c_{i}a_{i}/{\left(f_{i,j}\right)}^{3}$, which is greater than 0. $\partial^{2}U^{N}/\left(\partial f_{i,j}\partial f_{k,j}\right)=0$, $i,k\in N^{N}$, i≠k. The Hessian matrix of $U^{N}$ is positive definite. Hence, $U^{N}$ is convex. P1.3 can be solved by using the Lagrange multiplier method. The Lagrangian function is constructed as: (18)$$\begin{array}{c} L\left(f_{i,j},\lambda \right)=\sum_{i\in N^{N}}\left(\frac{a_{i}}{r_{i,1}}+\frac{2d_{i,1}}{c}+\frac{a_{i}}{r_{1,j}}+\frac{2l_{1,j}}{c}+\frac{c_{i}a_{i}}{f_{i,j}}\right)+\lambda \left(\sum_{i\in N^{N}}f_{i,j}-F_{j}\right), \end{array}$$
where λ is the Lagrange multiplier and λ≥0. The KKT conditions of P1.3 are listed as: (19)$$\begin{array}{c} \frac{\partial L\left(f_{i,j},\lambda \right)}{\partial f_{i,j}}=-\frac{c_{i}a_{i}}{{\left(f_{i,j}\right)}^{2}}+\lambda =0, \end{array}$$(20)$$\begin{array}{c} \frac{\partial L\left(f_{i,j},\lambda \right)}{\partial \lambda }=\sum_{i\in N^{N}}f_{i,j}-F_{j}=0, \end{array}$$(21)$$\begin{array}{c} \lambda \left(\sum_{i\in N^{N}}f_{i,j}-F_{j}\right)=0,\\ (8c)\;\mathrm{and}\;(8e), \end{array}$$(22)$$\begin{array}{c} \lambda \geq 0. \end{array}$$

Similar to solving P1.1, $f_{i,j}=\sqrt{c_{i}a_{i}/\lambda }$ is obtained from (Equation 19). Substituting it into (Equation 20), we can obtain $\sum_{i\in N^{N}}\sqrt{c_{i}a_{i}/\lambda }-F_{j}=0$, i.e., $\sqrt{\lambda }=\left(\sum_{i\in N}\sqrt{c_{i}a_{i}}\right)/F_{j}$. So $f_{i,j}=F_{j}\sqrt{c_{i}a_{i}}/\left(\sum_{i\in N^{N}}\sqrt{c_{i}a_{i}}\right)$. Given ([Disp-formula FD8e-sensors-23-00668]), we can obtain: (23)$$f_{i,j}^{*}=\min\left[\max\left(\frac{F_{j}\sqrt{c_{i}a_{i}}}{\sum_{i\in N^{N}}\sqrt{c_{i}a_{i}}},0\right),F_{j}\right].$$

By solving P1.1 and P1.3 above, we can obtain $f_{i,1}^{*}$ and $f_{i,j}^{*}$, respectively. By taking them back to P, P can be transformed into P2.
(24)$$\begin{array}{cc} P2: & \min_{\left\{y_{i,0},y_{i,1},y_{i,j}\right\}}\;\sum_{i\in N}\left[y_{i,0}\frac{c_{i}a_{i}}{f_{i,0}}+y_{i,1}\left(\frac{a_{i}}{r_{i,1}}+\frac{2d_{i,1}}{c}+\frac{c_{i}a_{i}}{f_{i,1}}\right)+\sum_{j\in M}y_{i,j}\left(\frac{a_{i}}{r_{i,1}}+\frac{2d_{i,1}}{c}+\frac{a_{i}}{r_{1,j}}+\frac{2l_{1,j}}{c}+\frac{c_{i}a_{i}}{f_{i,j}}\right)\right],\\ \; & s.t.\;(7f)\;\mathrm{and}\;(7g). \end{array}$$

Equation (Equation 24) is the objective function. P2 is an integer linear programming (ILP) problem, which is NP-hard. When the scale of a problem is large, it is difficult to obtain the optimal solution in polynomial time. ILP problems can be solved by intelligence optimization algorithms, one of which is the GWO algorithm. The GWO algorithm is widely used and has few parameters and is easy to implement [34].

According to the GWO algorithm [35], the grey wolf population is highly hierarchical and each wolf represents a feasible solution. The top rank wolf is α, i.e., the optimal solution. The second rank wolf is β, i.e., the suboptimal solution. The third rank wolf is δ, i.e., the third optimal solution. The remaining wolves are ω, which are the lowest rank and form a set $N^{\omega }$, i.e., the remaining solutions. The prey is the optimal solution to the problem. During each iteration, α, β, and δ command ω to encircle, hunt, and attack the prey to obtain the optimal solution.

We use $N^{W}$ to represent the number of grey wolves in the population. The wolf *l* is encoded as $\vec{X_{l}}=\left[x_{l,1},x_{l,2},x_{l,3},\ldots ,x_{l,N}\right]$, where $x_{l,i}$ is the task offloading decision of task *i* of wolf *l*. To be specific, $x_{l,i}=0$ represents local computing, $x_{l,i}=1$ means local satellite edge computing, $x_{l,i}=j$, j∈M denotes neighboring satellite *j* edge computing. After such coding, Equations ([Disp-formula FD7f-sensors-23-00668]) and ([Disp-formula FD7g-sensors-23-00668]) are satisfied. The grey wolves’ hunting of prey consists of three phases, i.e., encircling, hunting, and attacking.

During the encircling phase, the grey wolves scour the prey, and then approach and encircle it gradually. The distance between *l* and the prey *p* is denoted by $\vec{D}=|\vec{C}\cdot {\vec{X}}_{p}\left(t\right)-{\vec{X}}_{l}\left(t\right)|$, $\forall l\in N^{\omega }$, and the updating of ${\vec{X}}_{l}$ is ${\vec{X}}_{l}\left(t+1\right)={\vec{X}}_{p}\left(t\right)-\vec{A}\cdot \vec{D}$, where *t* is the number of iterations, ${\vec{X}}_{l}$ is the position of *l*, ${\vec{X}}_{p}$ is the position of *p*, and $\vec{A}$ and $\vec{C}$ are coefficient vectors. $\vec{A}=2\vec{a}\cdot {\vec{r}}_{1}-\vec{a}$, $\vec{C}=2\cdot {\vec{r}}_{2}$, where $\vec{a}$ is the convergence factor, which reduces from 2 to 0 linearly as *t* increases, $\vec{a}=2\cdot \left(1-t/t^{\max}\right)$. Both ${\vec{r}}_{1}$ and ${\vec{r}}_{2}$ are random vectors in $\left[0,1\right]$. $t^{\max}$ is the maximum number of iterations.

During the hunting phase, the grey wolves ω identify and track the location of prey according to the command of α, β, and δ. The distance between *l* and α, β, δ are expressed as ${\vec{D}}_{\alpha }=|{\vec{C}}_{1}\cdot {\vec{X}}_{\alpha }-{\vec{X}}_{l}|$, ${\vec{D}}_{\beta }=|{\vec{C}}_{2}\cdot {\vec{X}}_{\beta }-{\vec{X}}_{l}|$, and ${\vec{D}}_{\delta }=|{\vec{C}}_{3}\cdot {\vec{X}}_{\delta }-{\vec{X}}_{l}|$ respectively, where ${\vec{X}}_{\alpha }$, ${\vec{X}}_{\beta }$, and ${\vec{X}}_{\delta }$ are the positions of α, β, and δ in that order. ${\vec{C}}_{1}$, ${\vec{C}}_{2}$, and ${\vec{C}}_{3}$ are all random vectors. The step length and direction that *l* moves forward to α, β, and δ are ${\vec{X}}_{1}={\vec{X}}_{\alpha }-{\vec{A}}_{1}\cdot \left({\vec{D}}_{\alpha }\right)$, ${\vec{X}}_{2}={\vec{X}}_{\beta }-{\vec{A}}_{2}\cdot \left({\vec{D}}_{\beta }\right)$, and ${\vec{X}}_{3}={\vec{X}}_{\delta }-{\vec{A}}_{3}\cdot \left({\vec{D}}_{\delta }\right)$ respectively. The updating of ${\vec{X}}_{l}$ is ${\vec{X}}_{l}\left(t+1\right)=\left({\vec{X}}_{1}+{\vec{X}}_{2}+{\vec{X}}_{3}\right)/3$.

During the attacking phase, the prey stops moving and the grey wolves attack it. As the value of $\vec{a}$ decreases from 2 to 0 linearly, the value of $\vec{A}$ varies within the interval $\left[-|\vec{a}|,|\vec{a}|\right]$. When $|\vec{A}|<1$, the grey wolves the prey. When $|\vec{A}|>1$, the grey wolves move away from it to find a better one, which is conducive to jump out of the local optimum.

In addition to $\vec{A}$, another search coefficient, i.e., $\vec{C}$ also helps obtain the global optimal solution. $\vec{C}$ is composed of random values in $\left[0,2\right]$, which can provide random weights for the prey, so that the search of prey by grey wolves has randomness, thus helping to avoid falling into the local optimum.

The fitnesses of *l*, α, β, and δ are $U_{l}$, $U_{\alpha }$, $U_{\beta }$, and $U_{\delta }$, respectively. During each iteration, the GWO algorithm compares $U_{l}$ with $U_{\alpha }$, $U_{\beta }$, and $U_{\delta }$, and then updates ${\vec{X}}_{\alpha }$, ${\vec{X}}_{\beta }$, ${\vec{X}}_{\delta }$, $U_{\alpha }$, $U_{\beta }$, and $U_{\delta }$. When $t^{\max}$ is reached, ${\vec{X}}_{\alpha }$ is the optimal solution to the problem, and $U_{\alpha }$ is the optimal objective function value. According to ${\vec{X}}_{\alpha }$, $y_{i,0}^{*}$, $y_{i,1}^{*}$, and $y_{i,j}^{*}$ can be determined.

In this paper, we propose an inter-satellite cooperative task offloading scheme based on the GWO algorithm. We make some modifications to the GWO algorithm. Since ${\vec{X}}_{l}$ is a real number, which is inconsistent with the task offloading decision in integer form, we round it to convert the real number code into the integer code, so that the corresponding objective function value can be calculated and obtained.

The detailed step of the proposed joint offloading decision and resource allocation optimization scheme, which consists of a task offloading decision algorithm based on the GWO algorithm and a computing resource allocation algorithm based on the Lagrange multiplier method, and considers inter-satellite cooperation (GWOORAC), is expressed in Algorithm 1.

**Algorithm 1** GWOORAC
1: Initialize the grey wolf population, $\vec{a}$, $\vec{A}$, and $\vec{C}$. Set t=1. Set $t^{\max}$.

2: Obtain $f_{i,1}^{*}$ by (Equation 15), obtain $f_{i,j}^{*}$ by (Equation 23), thus calculating the fitness of each grey wolf. The top three wolves in fitness are identified as α, β, and δ, the other wolves are ω.

3: **while**
$1\leq t\leq t^{\max}$
**do**

4:   **for** each $l\in N^{\omega }$ **do**

5:     Update ${\vec{X}}_{l}$.

6:     Update $\vec{a}$, $\vec{A}$, and $\vec{C}$.

7:     For ${\vec{X}}_{l}$, obtain $f_{i,1}^{*}$ by (Equation 15), obtain $f_{i,j}^{*}$ by (Equation 23), thus calculating $U_{l}$.

8:     **if**
$U_{l}<U_{\alpha }$
**then**

9:        update ${\vec{X}}_{\alpha }={\vec{X}}_{l}$, $U_{\alpha }=U_{l}$.

10: **else if**
$U_{l}>U_{\alpha }$ and $U_{l}<U_{\beta }$
**then**

11:        update ${\vec{X}}_{\beta }={\vec{X}}_{l}$, $U_{\beta }=U_{l}$.

12: **else if**
$U_{l}>U_{\alpha }$ and $U_{l}>U_{\beta }$ and $U_{l}<U_{\delta }$
**then**

13:        update ${\vec{X}}_{\delta }={\vec{X}}_{l}$, $U_{\delta }=U_{l}$.

14:     **end if**

15:   **end for**

16:   t=t+1.

17: **end while**

18: **return**${\vec{X}}_{\alpha }$ and $U_{\alpha }$.


To sum up, the ${\vec{X}}_{\alpha }$ that is obtained finally is the optimal task offloading decision, i.e., $y_{i,0}^{*}$, $y_{i,1}^{*}$, and $y_{i,j}^{*}$. The corresponding $f_{i,1}^{*}$ and $f_{i,j}^{*}$ are the optimal computing resource allocation. $U_{\alpha }$ is the minimum task completion delay. The complexity of Algorithm 1 is $O(t^{\max}N^{W}N)$.

## 6. Simulation and Analysis

In this section, the delay is evaluated by performing our scheme and other baseline schemes. The results are also analyzed.

### 6.1. Simulation Parameters

The simulation parameters of this paper are shown in Table 1. The variables $a_{i}$ and $f_{i,0}$ follow random distribution within the intervals of [0.5, 5] Mbits and [1, 2] Gcycles/s, respectively. *K* is the Rice factor. The main references for simulation parameter settings are [16,24,25,27,30,31]. To validate our scheme, GWOORAC was compared with other baseline schemes described below. The first scheme uses the GWO algorithm to determine the offloading decision, allocates the resources randomly, and considers inter-satellite cooperation (GWORandRAC). The second scheme obtains the offloading decision randomly, optimizes the resource allocation, and considers inter-satellite cooperation (RandORAC). The third scheme uses the GWO algorithm to make the offloading decision, optimizes the resource allocation, and takes no account of inter-satellite cooperation (GWOORANC). The fourth scheme only has local computing, and the rest is the same as our scheme (OLC). The fifth scheme uses the particle swarm optimization (PSO) algorithm to determine the offloading decision, optimizes the resource allocation, and considers inter-satellite cooperation (PSOORAC). The number of particles is 20. The maximum number of iterations is 200. The inertia weight is 0.8 and both learning factors are 1.5. All simulation results were obtained by averaging 300 times simulations.

### 6.2. Simulation Results and Analysis

As shown in Figure 3, as the number of IoT devices, i.e., *N*, increases, the delay of all schemes increases gradually. GWOORAC has the lowest delay, and the more IoT devices, the more the delay reduction. Compared with GWORandRAC, the delay is relatively close when N=10. However, when N>10, the delay reduction increases notably, which demonstrates that our optimal resource allocation is effective. The delay of GWOORAC is lower than that of RandORAC. Moreover, the delay reduction increases notably when N>15, which proves the effectiveness of the GWO algorithm. Compared with GWOORANC, there is little difference in the delay when N≤25. By contrast, when N>25, the difference between the delay of GWOORAC and that of GWOORANC increases gradually, which indicates that the inter-satellite cooperation is of great significance in reducing the delay. The computing resources of MEC servers are much larger than those of IoT devices, and each task can be allocated more computing resources than the computing capacity of IoT devices when tasks are offloaded to satellites for computing; thus, the delay of OLC is much higher than that of GWOORAC. Moreover, although the delay increases as *N* increases, the increasing speed of GWOORAC is much smaller than that of OLC.

Figure 4 shows that GWOORAC has the lowest delay. As the number of satellites, i.e., *M* increases, the delay of GWOORAC, GWORandRAC, and RandORAC decreases gradually, because more neighboring satellites bring more computing resources, and more computing resources are available for each offloaded task. Neither GWOORANC nor OLC takes into account neighboring satellites, so their delay almost does not change with the variation of *M*. Due to the optimal resource allocation, the delay of GWOORAC is much lower than that of GWORandRAC. The delay of GWOORAC is much lower than that of RandORAC, which indicates that the GWO algorithm is appropriate to solve the problem. When M≥2, the delay of GWOORAC is lower than that of GWOORANC, and the gap gets larger, which demonstrates that the inter-satellite cooperation reduces the delay. Compared with OLC, the delay of GWOORAC is much lower, and the delay reduction becomes larger as *M* increases, because a large number of computing resources can be allocated to each offloaded task, far more than the computing capacity of IoT devices.

As Figure 5 shows, the delay of all schemes except OLC shows downward trends as the computing resources of each satellite, i.e., $F_{1}$ and $F_{j}$, increase. The reason is that the more computing resources a satellite has, the more computing resources are allocated to offloaded tasks, which reduces the delay of computing tasks. OLC does not consider computation offloading, so its delay is not affected by $F_{1}$ and $F_{j}$. Due to reasonable resource allocation, the delay of GWOORAC is much lower than that of GWORandRAC. The delay of GWOORAC is lower than that of RandORAC, which indicates that the GWO algorithm can obtain an appropriate task offloading decision, thereby reducing the delay. The delay of GWOORAC is lower than that of GWOORANC, which proves that the inter-satellite cooperation reduces the delay effectively. The delay of GWOORAC is much lower than that of OLC, and the delay reduction becomes larger gradually. This demonstrates that a combination of suitable task offloading decisions and reasonable computing resource allocation can reduce the delay significantly.

As shown in Figure 6, the delay of all schemes shows upward trends as the computation intensity, i.e., $c_{i}$, increases, because of the increased task computation delay. The delay of GWOORAC is much lower than that of GWORandRAC due to the optimal computing resource allocation. The delay of RandORAC is higher than that of GWOORAC, which validates that the GWO algorithm can obtain the proper task offloading decision, thus reducing the delay. Thanks to the inter-satellite cooperation, the delay of GWOORAC is lower than that of GWOORANC. The computing capacity of IoT devices is weaker than that of MEC servers, so the delay of OLC is higher than that of GWOORAC, and its growth speed is much faster than that of GWOORAC. When $c_{i}\leq 400$ cycles/bit, the delay of GWOORAC and other schemes is fairly close. When $c_{i}>400$ cycles/bit, the gap becomes larger, which validates that both the appropriate task offloading decision and the proper computing resource allocation are necessary.

As shown in Figure 7, the data size of each task, i.e., $a_{i}$ is assumed to be the same. As $a_{i}$ increases, the delay of all schemes increases. The reason is that the data size is bigger, and the corresponding task transmission delay and task migration delay are higher. Additionally, the computing resources required by a task are proportional to its data size, which means that big data sizes bring high task computation delay. Compared with GWORandRAC, the delay of GWOORAC is much lower due to the optimal resource allocation. As can be seen from the fact that the delay of GWOORAC is lower than that of RandORAC, the task offloading decision obtained by the GWO algorithm can reduce the delay. The gap between the delay of GWOORAC and that of GWOORANC shows the contribution of the inter-satellite cooperation. The delay of OLC is much higher than that of GWOORAC, which validates the effectiveness of GWOORAC.

As Figure 8 shows, the delay of both schemes increases with the increase of *N*. The delay of GWOORAC is lower than that of PSOORAC. This is because the values of the parameters, such as $\vec{A}$, are adaptive, allowing for a better balance between exploration and exploitation, thus more effectively avoiding falling into the local optimum.

## 7. Conclusions

In this paper, we investigate inter-satellite cooperative task offloading and resource allocation in an MEC-enabled STN. The task completion delay optimization problem for all IoT devices when the tasks are indivisible is formulated and decomposed. Then, we propose a joint offloading decision and resource allocation optimization scheme, which consists of a task offloading decision algorithm based on the GWO algorithm and a computing resource allocation algorithm based on the Lagrange multiplier method, thus obtaining the optimal task offloading decision and allocating optimal computing resources of the local satellite’s MEC server and neighboring satellites’ MEC servers. Simulation results validate that the proposed scheme performs better than the other baseline schemes in the following cases: the number of IoT devices is large, the number of satellites is large, each satellite has numerous computing resources, the computation intensity is high, or the data size is big. Much work remains to be done in the future. For example, we will investigate inter-satellite cooperation, where multiple satellites with MEC servers can simultaneously receive tasks offloaded by IoT devices and assist IoT devices in computing tasks in areas not covered by terrestrial networks.

## Figures and Tables

**Figure 1 sensors-23-00668-f001:**
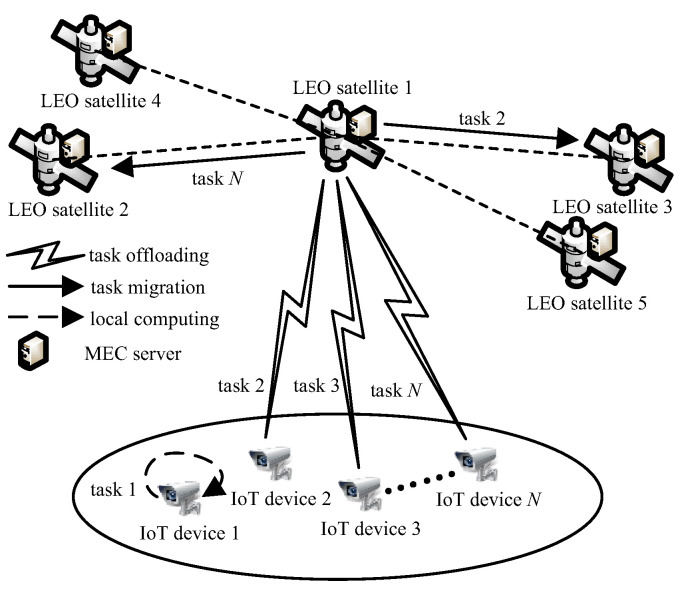
Network model.

**Figure 2 sensors-23-00668-f002:**
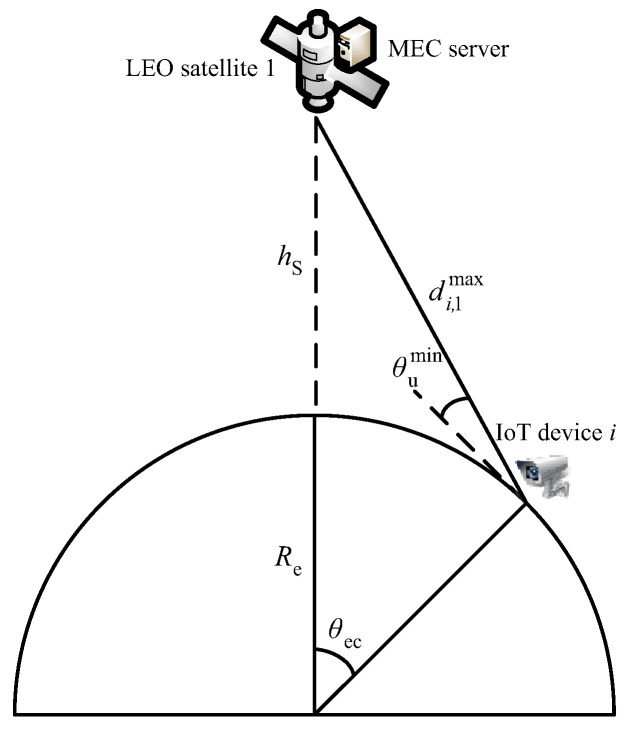
Illustration of $d_{i,1}^{\max}$.

**Figure 3 sensors-23-00668-f003:**
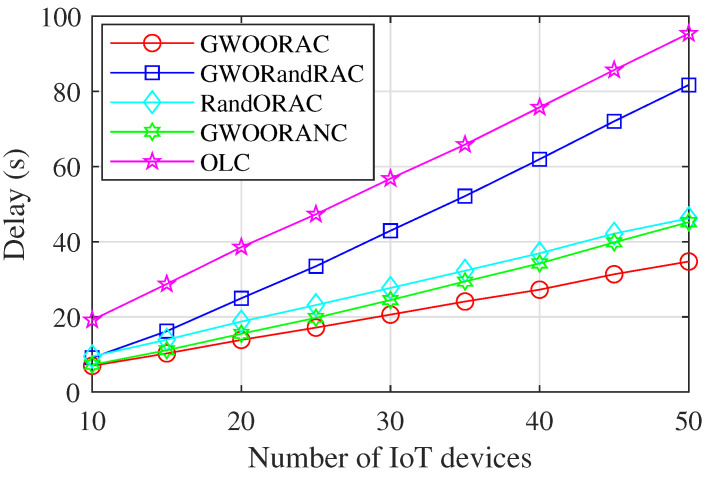
Delay vs. number of IoT devices.

**Figure 4 sensors-23-00668-f004:**
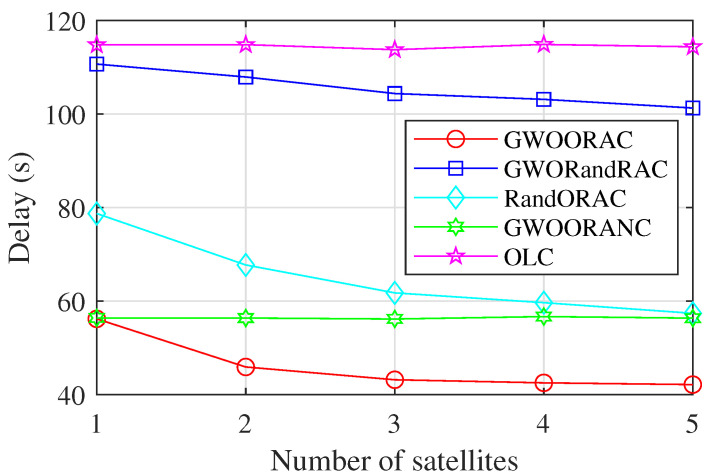
Delay vs. number of satellites.

**Figure 5 sensors-23-00668-f005:**
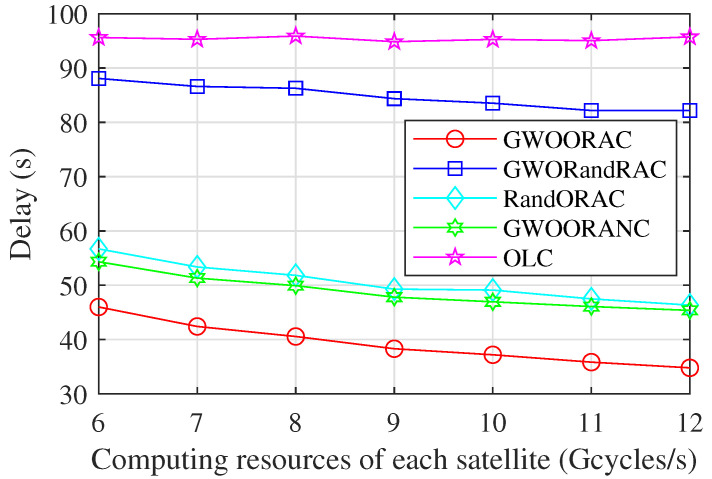
Delay vs. computing resources of each satellite.

**Figure 6 sensors-23-00668-f006:**
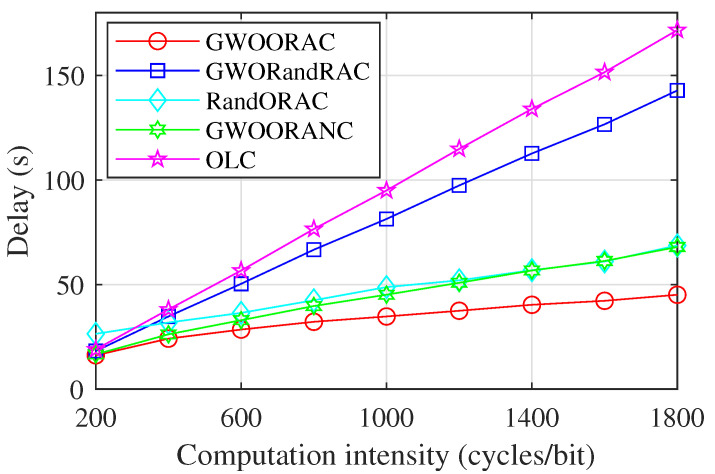
Delay vs. computation intensity.

**Figure 7 sensors-23-00668-f007:**
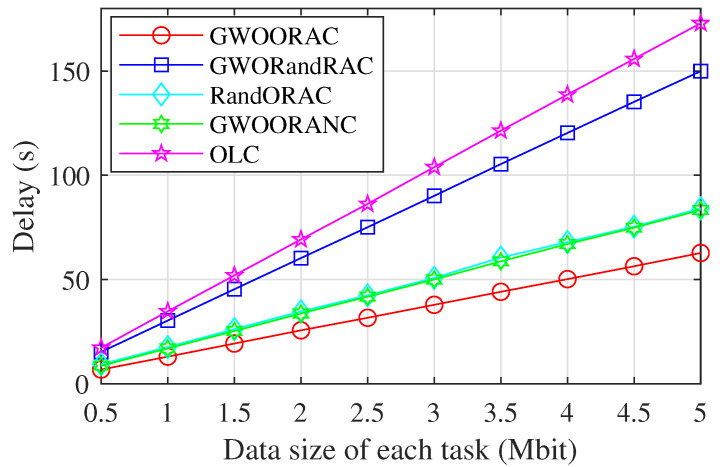
Delay vs. data size of each task.

**Figure 8 sensors-23-00668-f008:**
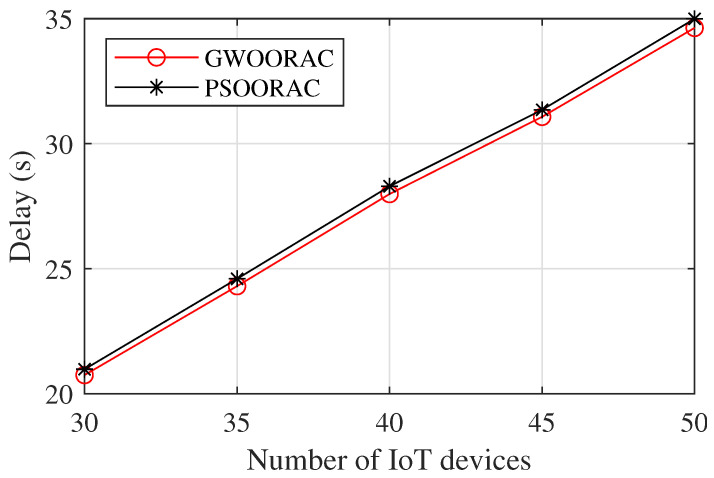
Delay vs. number of IoT devices.

**Table 1 sensors-23-00668-t001:** Simulation parameters.

Parameter	Value	Parameter	Value
*N*	50	*M*	5
$c_{i}$	1000 cycles/bit	$a_{i}$	[0.5, 5] Mbits
$f_{i,0}$	[1, 2] Gcycles/s	$p_{i}$	2 W
$g_{i}$	43.3 dBi	$B_{i,1}$	100 MHz
$B_{1,j}$	100 MHz	*P*	50 W
$F_{1}$	12 Gcycles/s	$F_{j}$	12 Gcycles/s
$D_{\mathrm{TA}}$	2.2 m	$N_{0_{i,1}}$	−203 dBm/Hz
$N_{0_{1,j}}$	−203 dBm/Hz	$R_{e}$	6371 km
$\theta_{\min}^{U}$	$16^{\circ }$	lat	$60^{\circ }$
$h_{s}$	780 km	*c*	$3\times 10^{8}$ m/s
*f*	30 GHz	*K*	7
*I*	6	*J*	11
$N^{W}$	20	$t^{\max}$	200

## Data Availability

Not applicable.

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
