# Peer review of "Inter-Satellite Cooperative Offloading Decision and Resource Allocation in Mobile Edge Computing-Enabled Satellite–Terrestrial Networks"

_sensors, 2023, doi:10.3390/s23020668_

Round 1

Reviewer 1 Report

The authors of this manuscript aim to examine the inter-satellite cooperative task offloading and resource allocation in the mobile edge computing (MEC)-enabled satellite-terrestrial networks (STNs). They formulate the task completion delay optimization problem for all IoT devices when tasks are. Next, they propose a joint offloading decision and resource allocation optimization algorithm. This algorithm makes use of a gray wolf optimization (GWO) and a computing resource allocation algorithm based on Lagrange multiplier method. This allows them to obtain the optimal task offloading decision and to allocate the optimal computing resources of the local satellite’s MEC  server and neighbor satellites’ MEC servers. Their numerical results confirm that their proposed algorithm outperforms counterpart algorithms from the literature, in terms of different comparisons that involve delay and number of satellites, for example. Overall, the manuscript is well-written. Sometimes the authors make a mistake when using (or forgetting to use) the definite article “the”. The introduction and related works sections provide a good context for the proposed ideas, as well as present the state-of-the-art in a good manner. The references are recent and adequate. The system model is very presented. This is aided with the related figures and descriptions. The problem formulation is also well-written and explained. This is aided with the provision of the needed mathematical formulae. The GWO algorithm is also thoroughly explained in this section. The simulation and analysis section provides a clear description of the simulation parameters employed and the computed parameters. This is well-presented in plots that compare the performance of the proposed ideas with those of their counterparts from the literature. The proposed algorithm is shown to outperform in every aspect. The final section concludes the manuscript, summarizing it and reiterating the major achievement of the work.

Minor language revision is required. Examples of mistakes include lines 28, 35, 44, 278.

Reviewer 2 Report

The article presents the Grey Wolf Optimizer algorithm for resource allocation for mobile edge computing. The simulated results support the proposed algorithm. 

I would like to suggest authors carefully check for grammatical mistakes.

Reviewer 3 Report

The manuscritp was well written and organized. It is better to conduct real experiments to validate the proposed algorithms.

Reviewer 4 Report

A task offloading decision algorithm based on GWO and a computing resource allocation algorithm based on Lagrange multiplier method under mobile edge computing-enabled satellite-terrestrial networks are proposed, respectively. By comparing with other schemes, the experimental results show the excellent performance of the proposed GWOORAC algorithm. However, the authors are encouraged to revise their paper according to the following comments.

1. In related work, it is suggested that the authors add some introductions to the mainstream algorithms for computation offloading and resource allocation, and analyze their advantages and disadvantages.

2. The authors should add some introduction to the GWO algorithm and analyze its advantages in this paper.

3. The author should explain the mechanism of task generation and the order in which tasks are processed.

4. The author should explain the rules of motion of satellites and how to ensure that neighboring satellites are always within the communication range of the local satellite.

5. In the simulation experiments, the authors should add some comparison results of other intelligent optimization algorithms.

6. The author should discuss the limitations and future research work.
